# Stereotactic Body Radiotherapy for Renal Cell Carcinoma in Patients with Von Hippel–Lindau Disease—Results of a Prospective Trial

**DOI:** 10.3390/cancers14205069

**Published:** 2022-10-17

**Authors:** Simon Kirste, Alexander Rühle, Stefan Zschiedrich, Wolfgang Schultze-Seemann, Cordula A. Jilg, Elke Neumann-Haefelin, Simon S. Lo, Anca-Ligia Grosu, Emily Kim

**Affiliations:** 1Department of Radiation Oncology, Medical Center—University of Freiburg, Faculty of Medicine, 79106 Freiburg, Germany; 2German Cancer Consortium (DKTK) Partner Site Freiburg, German Cancer Research Center (dkfz), 69120 Heidelberg, Germany; 3Renal Division, Department of Internal Medicine, Bürgerspital Solothurn, 4500 Solothurn, Switzerland; 4Faculty of Medicine, University of Freiburg, 79106 Freiburg, Germany; 5Department of Urology, Medical Center—University of Freiburg, Faculty of Medicine, 79106 Freiburg, Germany; 6Renal Division, Department of Medicine, Medical Center—University of Freiburg, Faculty of Medicine, 79106 Freiburg, Germany; 7Department of Radiation Oncology, University of Washington School of Medicine, Seattle, WA 98195, USA

**Keywords:** stereotactic body radiotherapy, SBRT, stereotactic ablative radiotherapy, SABR, renal cell carcinoma, von Hippel–Lindau disease

## Abstract

**Simple Summary:**

Clear cell renal cell carcinoma (ccRCC) frequently occurs in patients with von Hippel–Lindau disease and is a leading cause of mortality in patients with this hereditary disorder. Partial nephrectomy, which is the standard treatment, is often complicated by multilocular tumor occurrence in both kidneys requiring repeated surgeries. Consequently, nephron-sparing resections become increasingly difficult ultimately leading to chronic kidney failure. In these patients or in patients who refuse surgery, alternative treatment approaches are needed. In this study, we investigated the outcome and toxicities especially for the kidney after stereotactic body radiotherapy (SBRT). We could demonstrate that SBRT in this highly vulnerable group of patients is feasible without any high-grade adverse events in the long-term and results in excellent local control at the site of treatment. Efficacy of SBRT is already proven for other anatomic sites and it could represent a valuable, non-invasive treatment option for ccRCC as well, especially for patients who are extremely vulnerable to any kidney injury. Further prospective trials evaluating SBRT for localized ccRCC are necessary to verify the promising findings and to examine its role as an alternative to surgery in inoperable patients.

**Abstract:**

Von Hippel–Lindau disease (VHL) is a hereditary disorder associated with malignant tumors including clear cell renal cell carcinoma (ccRCC). Partial nephrectomy is complicated by multilocular tumor occurrence and a high recurrence rate. The aim of this study was to evaluate the potential of stereotactic body radiotherapy (SBRT) as an alternative treatment approach in VHL patients with multiple ccRCC. Patients with VHL and a diagnosis of ccRCC were enrolled. SBRT was conducted using five fractions of 10 Gy or eight fractions of 7.5 Gy. The primary endpoint was local control (LC). Secondary endpoints included alteration of renal function and adverse events. Seven patients with a total of eight treated lesions were enrolled. Median age was 44 years. Five patients exhibited multiple bilateral kidney cysts in addition to ccRCC. Three patients underwent at least one partial nephrectomy in the past. After a median follow-up of 43 months, 2-year LC was 100%, while 2-year CSS, 2-year PFS and 2-year OS was 100%, 85.7% and 85.7%, respectively. SBRT was very well tolerated with no acute or chronic toxicities grade ≥ 2. Mean estimated glomerular filtration rate (eGFR) at baseline was 83.7 ± 13.0 mL/min/1.73 m^2^, which decreased to 76.6 ± 8.0 mL/min/1.73 m^2^ after 1 year. Although the sample size was small, SBRT resulted in an excellent LC rate and was very well tolerated with preservation of kidney function in patients with multiple renal lesions and cysts.

## 1. Introduction

Von Hippel–Lindau (VHL) disease is an autosomal dominant disorder caused by mutations in the VHL tumor-suppressor gene, located on chromosome 3p25-26 [1,2]. The incidence is about one in 36,000 livebirths [3]. The disease was first described at the beginning of the 20th century by the German ophthalmologist Eugen von Hippel and the Swedish pathologist Arvid Lindau. They studied two cases of retinal angiomas and noted an association between cerebellar hemangioblastomas, retinal angiomas, and other visceral tumors [4,5]. Individuals with VHL disease are at risk of developing various benign and malignant tumors of the central nervous system, eyes, adrenal glands, pancreas, reproductive adnexal organs, and kidney for which clear cell renal cell carcinoma (ccRCC) is a frequent malignancy in VHL patients, often presenting as multifocal and bilateral tumor in these patients [6]. The lifetime risk of ccRCC in VHL patients is nearly 70% and is a leading cause of mortality [7,8].

Both hereditary and sporadic ccRCCs are characterized by mutations in the VHL gene and a subsequent loss of heterozygosity [8]. The lack of functional VHL tumor-suppressor protein prevents the ubiquitination and degradation of hypoxia-inducible factor α (HIFα) subunits, leading to abnormal accumulation of this transcription factor [9]. Together with HIF1β, HIFα subunits bind to hypoxia-response elements to activate genes encoding vascular endothelial growth factor (VEGF), platelet-derived growth factor (PDGF), erythropoietin (EPO) and transforming growth factor alpha (TGF-α), activating cellular programs such as angiogenesis, cell cycle, and cell proliferation, as well as metabolic changes. Furthermore, VHL inactivation was found to result in epithelial-mesenchymal transition via the NF-κB pathway [10].

The overriding goal of therapy for ccRCC in VHL patients is to detect and treat tumors before progression to metastatic disease, and to preserve kidney function. The current treatment standard is active surveillance by serial imaging, followed by nephron-sparing surgery for tumors reaching a certain size, which puts the patients at a higher risk to develop metastasis [11]. Unlike sporadic ccRCC, ccRCC in individuals with VHL disease is often multicentric, bilateral and prone to recurrence due to underlying microscopic disease. Repeated surgical interventions for new ccRCCs increase the risk of developing end-stage renal insufficiency, requiring dialysis and kidney transplantation. The need to manage serial small tumors has led to the use of non-surgical, less invasive, local ablative treatment techniques such as radiofrequency ablation (RFA), microwave ablation (MWA) and cryoablation [12,13]. These techniques, which are limited to treat a maximum tumor size of 2 cm, are invasive with a substantial risk of hemorrhage and infection, thus typically limited to small renal masses located away from the ureter and vascular structures because of the risk of heat sink effects, stricture and/or fistula development [14]. Long-term efficacy is difficult to assess due to small study sizes and limited follow-up.

Another ablative treatment modality is stereotactic body radiotherapy (SBRT) which uses high individual radiation doses in few fractions to eradicate tumor cells. SBRT is a non-invasive technique already routinely used for other primary tumors and metastases such as early stage lung cancer, primary liver tumors, and liver, lung and spinal metastases [15]. Data on the use of SBRT for primary RCC have emerged showing excellent local control (LC) rates of 84–100% at 2 years [16], although there are few prospective data [17,18]. Compared to local ablative techniques, the use of SBRT is less constraint by tumor proximity to renal hilum or by tumor size.

However, there are yet no studies that have evaluated the feasibility, safety, and efficacy of SBRT for ccRCC in patients with underlying VHL disease. This prospective phase II trial therefore aimed to evaluate the feasibility and outcome of SBRT for the treatment of ccRCC in this special patient population.

## 2. Materials and Methods

### 2.1. Patient Population

In the Medical Center—University of Freiburg, patients with a diagnosis of VHL are routinely monitored by a specialized VHL outpatient clinic. In case of a newly diagnosed or growing renal lesion, patients were presented for evaluation of study inclusion. Inclusion criteria included patients with a genetic diagnosis of VHL and a progressive renal tumor with ≥1.5 cm in diameter and typical features of RCC on MRI and/or CT. A biopsy was not mandatory as most of the patients already had a previous histologic confirmation of RCC. Patients with previously treated renal cell carcinoma by surgery or other local therapies were permitted. Chronic Kidney Disease (CKD) stage IV or worse or previous radiotherapy were exclusion criteria. Tumor size was not an exclusion criterion. All patients were discussed with surgeons, urologists, nephrologists and radiologists at a multidisciplinary tumor board. Mostly patients with RCC and multiple bilateral cysts or patients in which a surgical, nephron-sparing approach was technically not feasible or patients who refused surgery were included in the study. The study was approved by the institutional review board (ref. num. 266/15), and every patient had to give written informed consent before study inclusion.

### 2.2. Pre-Treatment Evaluations

Magnetic resonance imaging (MRI) scans including diffusion-weighted (DWI) and dynamic contrast-enhanced (DCE) MRI sequences with and without contrast were used to identify progressive lesions. Tumor size and volume as well as pretreatment growth rate of the treated lesion were measured on MRI. For SBRT planning purposes, every patient received a planning MRI within 2 weeks before start of treatment.

To examine kidney function for both sides separately, a pre-treatment renal nuclear scintigraphy with 102 MBq 99mTc-MAG3 was performed on every patient. The estimated glomerular filtration rate (eGFR) was calculated using the CKD-EPI formula.

The age-adjusted Charlson Comorbidity Score was calculated for all patients [19]. Based on the pre-treatment MRI scans the R.E.N.A.L. nephrometry score was determined. The score ranges continuously from 4 to 12. The complexity of renal masses according to R.E.N.A.L. score are deemed low in the range of 4 to 6, moderate in the range of 7 to 9 and high in the range of 10 to 12 [20].

### 2.3. Immobilization, Planning and Delivery of Stereotactic Body Radiotherapy

All patients underwent a planning CT in the supine position using a dedicated SBRT vacuum cushion and abdominal compression for immobilization and motion control. They underwent a free-breathing (FB; kVp = 120 kV, mAs = 350, slice thickness of 1 mm) and a respiration-correlated 4D-CT (kVp = 120 kV, mAs = 350, slice thickness of 2 mm) with a Brilliance CT BigBore Oncology computed tomography (CT) scanner (Philipps, Germany), using a belt for respiratory signal acquisition. Acquired images were reconstructed in different respiratory phases (0% inhalation to 100% inhalation in 10% increments).

The gross tumor volume (GTV) was contoured on the baseline planning CT and the co-registered pre-treatment MRI. An internal target volume (ITV) was defined by drawing the gross tumor volume (GTV) in all respiratory phases. The planning target volume (PTV) was defined by applying a 4 mm isotropic expansion to the ITV (Figure 1). The organ at risk that needed to be considered in most cases was the small bowel. A priority in the physical planning process was sparing of healthy renal parenchymal of the treated kidney and the avoidance of the contralateral kidney. All patients were treated using a Varian TrueBeam LINAC (6 and 15 MV photons). Prior to each treatment, a fb-cbCT was acquired using on-board imaging system. Optical surface imaging was used for initial setup, as well as tracking during treatment delivery. The median duration of one fraction was approximately 20 min. With the aim of maximum renal parenchyma sparing, radiotherapy was delivered using a volumetric modulated arc therapy (VMAT) with 2–5 arcs using photon energies of 6 or 18 MeV. Five fractions of 10 Gy were prescribed to the 95% isodose every other day. If predefined constraints for organ at risks could not be reached, prescription dose was reduced to 8 fractions of 7.5 Gy. Peak dose within the PTV was ≥120%.

### 2.4. Response Evaluation and Follow-Up

Tumor volume as well as kidney volume were measured using a segmentation tool program (Eclipse). Volumes were manually contoured on cross-sectional contrast-enhanced CT images of arterial and venous phases and T1-Gadolinium enhanced MRI sequences. The 3D volumes were automatically calculated afterwards.

All patients were monitored weekly during therapy. During follow-up, patients were routinely examined every 3 to 6 months. At each follow-up visit, patients had a medical history, physical examination, and imaging of the abdomen by MRI including DWI and DCE sequences. Toxicity was monitored and scored at each follow up visit.

Response evaluation was done on MRI scans according to the Response Evaluation Criteria In Solid Tumors version (RECIST) v1.1 [21]. Additionally, absolute change in tumor size from baseline to the respective time point was calculated. To compute tumor volume from the baseline planning CT scan and baseline MRI as well as from each MRI on follow up visits, the tumor was considered to have an ellipsoid shape, thus allowing calculating the volume according to the following equation: π/6 × height × length × width. Change in tumor volume compared to baseline MRI was calculated for each follow-up MRI.

Global and regional kidney function was measured by serum creatinine assessments at every follow-up visit and renal nuclear scintigraphy was done at baseline and at 1 year after treatment. The eGFR was calculated at each follow-up visit using the CKD-EPI formula. Treatment-related toxicities were defined using Common Terminology Criteria for Adverse Events (CTCAE) v4.0.

### 2.5. Statistical Analysis

LC, cancer-specific survival (CSS), progression-free-survival (PFS) and overall survival (OS) were determined following Kaplan–Meier analyses. Local relapse was defined as recurrence of the treated lesion using RECIST v1.1 criteria. PFS was defined as the time interval between start of SBRT until local recurrence, locoregional development of a novel ccRCC, distant progression or death from any cause. OS was considered as time interval between start of SBRT and death. Patients were censored at the last follow-up visit. Median follow-up time was calculated based on the reverse Kaplan–Meier method. IBM SPSS Statistics software version 25 (IBM, Armonk, NY, USA) and GraphPad Prism software version 8 (GraphPad Software, San Diego, CA, USA) were used for statistical analyses.

## 3. Results

### 3.1. Patient and Treatment Characteristics

Between February 2016 and June 2018, 7 patients treated for 8 lesions were enrolled in this prospective phase II study. Median age was 44 years (range, 36–56 years) and 4 patients (57.1%) were female (Table 1). The body mass index (BMI) was 23.3 (20.3–25.8). All patients had VHL disease and a history of multiple surgeries (e.g., resection of spinal cysts, adrenalectomy). Eastern Cooperative Oncology Group (ECOG) status was 0 in four patients (57.1%) and 1 in three patients. The age-adjusted Charlson Comorbiditiy Index was 2 in six and 3 in one patient. Six patients had a solitary ccRCC, and one patient was treated for two lesions in the left kidney. ccRCCs were right-sided in three patients (42.9%) and left-sided in four patients (57.1%). The median maximum tumor diameter and total tumor volume were 2.8 cm (range, 1.9–3.5 cm) and 8.6 cc (range, 2.9–16.8 cc) at the time of SBRT. Typical for VHL patients, kidneys were affected by multiple bilateral cysts in five patients (71.4%). Three patients (42.9%) underwent at least one partial nephrectomy before study inclusion. The Renal Nephrometry Score which describes tumor complexity was a median of 8.5 (4–10) with a low complexity score in 2, intermediate complexity score in 4 and high complexity score in 2 patients. The median relative cortical tubular function of the treated kidney was 45.0% (range, 21.0–55.0%) as measured by renal nuclear scintigraphy with 102 MBq 99mTc-MAG3.

Treatment was performed with 5 × 10 Gy in seven lesions and 8 × 7.5 Gy in one lesion due to proximity to small bowel (Table 2). Median PTV size was 27.0 cc (range, 15.6–44.0 cc). Kidney movement by respiration was assessed by 4D-CT, increasing the GTV by a factor of 1.4 resulting in a median ITV of 13.2 cc (range, 5.6–22.1 cc). The median dose to the PTV was 51.0 Gy (range, 46–57 Gy) and the maximum dose was 57.3 Gy (range, 53.5–63) in lesions receiving 5 fractions and 67.3 Gy and 76 Gy in the lesion receiving 8 fractions, corresponding to biological effective doses of 103.0 Gy and 123.8 Gy, respectively.

### 3.2. Outcomes and Toxicity

All patients were treated as initially planned, and there was no interruption of therapy. After a median follow-up time of 43 months (range, 18–54 months), LC rate was 100% after 1 year, 2 years and 3 years (Figure 2). Three lesions were classified as partial remission (PR) and 5 lesions as stable disease (SD) according to RECIST v1.1 criteria. 2-year CSS, 2-year PFS and 2-year OS was 100.0%, 85.7% and 85.7%, respectively. At the date of analysis, six patients were still alive without signs of local progression, whereas one patient died from a bilateral pneumonia 18 months after SBRT.

Radiological response of the treated lesions was slow over the follow-up period with most lesions being stable or showing a small decrease in size. Contrast enhancement diminished over time with lesions showing necrotic areas in the middle of the lesion. Figure 3A displays the relative change in tumour volume at 12 months after SBRT in the individual lesions. A total of 3 lesions (37.5%) exhibited a decrease of ≥30% in their tumor volumes, therefore classified as PR according to RECIST v1.1. One lesion was found to show an increase of 30 % at 9 months after SBRT with pronounced central necrosis and was regredient at further follow-up.

Absolute changes in tumor volume are shown in Figure 3B. At 12 months prior to SBRT, median tumor size and volume amounted to 2.1 cm (range, 1.5–3.5 cm) and 3.6 cc (range, 0.8–16.8 cc), respectively. Tumor volume was slightly decreasing 1 year after treatment until last follow-up in all patients.

Treatment was very well tolerated with no acute toxicities grade ≥2 (Table 3). Only one patient experienced mild flank pain at the site of treatment, which began after the third fraction. Notably, no acute gastrointestinal toxicities, nor any late toxicities were observed.

### 3.3. Effect on Kidney Function

Concerning global renal function, mean (± standard deviation) serum creatinine levels were 0.95 ± 0.16 mg/mL at the start of SBRT and 0.93 ± 0.18 mg/mL at the end of treatment. At 6, 12 and 24 months after SBRT, mean serum creatinine levels were 1.00 ± 0.16 mg/dL, 1.04 ± 0.16 mg/mL and 0.98 ± 0.11 mg/mL, respectively (Figure 4A).

The mean eGFR was 83.7 ± 13.0 mL/min/1.73 m^2^ at baseline and 86.0 ± 10.7 mL/min/1.73 m^2^ at the end of SBRT (Figure 4B). At 6, 12 and 24 months after treatment, eGFR was 78.4 ± 14.2 mL/min/1.73 m^2^, 76.6 ± 8.0 mL/min/1.73 m^2^ and 78.5 ± 14.9 mL/min/1.73 m^2^, respectively. The maximum eGFR decrease was 21.1 mL/min/1.73 m^2^ during the first 24 months after SBRT in a 56-year-old female patient with two treated RCCs (Figure 4C). Importantly, no patient required dialysis. Split renal function in the irradiated kidney declined from a mean 45.0% at baseline to 41.8% at 1 year after treatment. To compensate for this effect the split renal function of the non-irradiated kidney increased from 55% to 58.2%.

## 4. Discussion

To the best of our knowledge, this is the first prospective study that investigated the feasibility, safety, and efficacy of SBRT in the treatment of hereditary ccRCC in VHL patients. It was demonstrated that SBRT resulted in a LC rate of 100% after a median follow-up time of 43 months although results must be interpreted with caution as the patient number is limited. Importantly, there were no acute or chronic toxicities grade ≥2, and no patient required dialysis within the follow-up.

The treatment of primary RCC with SBRT is still a relatively new field. Radiotherapy using conventionally fractionated doses did not show acceptable results for RCC in the past [22,23]. In contrast, SBRT resulted in excellent LC rates in several studies and is being adopted for inoperable patients in SBRT experienced centres [24]. So far, there are few prospective trials that have evaluated the role of SBRT for RCC [25,26,27,28,29]. A meta-analysis with 26 studies including 372 patients has shown excellent results with a LC of 97.2% and grade 3–4 toxicity of only 1.5% [16].

VHL patients differ greatly from patients affected by sporadic ccRCC which were included in contemporary studies of SBRT for ccRCC so far. VHL patients are significantly younger, often affected by bilateral tumors and multiple cysts. The average age of first diagnosis of ccRCC is 44 years in VHL patients versus 64 years in patients with sporadic ccRCC [30]. VHL patients are much more likely to be diagnosed with multiple and bilateral disease, originating from microscopic precursor lesions present in both kidneys [31]. The number of non-malignant cysts lined with clear cells in an average VHL kidney was estimated to be 1100, and the number of clear cell renal neoplasms (solid and cystic) to be 600 [32]. Multiple renal cysts can additionally affect the kidney function. For these reasons, nephron-sparing approaches are heavily needed in this patient cohort, as VHL patients often require multiple resections with a high consecutive risk of chronic kidney function loss.

Other local ablative treatment techniques that could be an alternative to resection are RFA, MWA and cryoablation. In line with the studies regarding SBRT for RCC, most studies deliberately excluded patients with hereditary syndromes, which is why data for RFA in VHL patients is very limited [14,33,34]. In a study by Allasia et al., 9 patients with VHL disease were treated with RFA [14]. In this study, RFA could be safely performed, and the disease was controlled in 6 patients. Lesions were relatively small with a diameter of 2.5 cm and follow-up was too short to draw any conclusions on renal function after treatment. In another study, 11 patients with VHL were treated with RFA for renal tumors. Overall 8 patients could be treated successfully but two major complications including arteriovenous fistula and ureteral perforation occurred, rendering the treatment not suitable for VHL patients [34].

In comparison with RFA or MWA, SBRT exhibits several advantages: There are no limitations for size nor proximity to large vessels. Furthermore, SBRT is a non-invasive technique, avoiding the need for anaesthesia. Additionally, SBRT seems to be more cost effective compared to RFA as demonstrated by an analysis conducted in the Canadian healthcare system [35].

In the present study, we did not observe any acute or late toxicities grade ≥2 after SBRT. Mild side effects included fatigue and local muscle pain. Toxicity rates for SBRT in primary RCC are generally very favourable. The toxicity results in our trial compare favourably with the published results for non-VHL associated RCC. In the prospective trial by Siva et al., 21% of patients experienced grade 2 toxicity, mostly fatigue and nausea, and 58% of patients experienced grade 1 toxicity, mostly comprised of chest wall pain and fatigue [29]. Even when using single doses of 25 Gy, as in the largest radiosurgery trial by Staehler et al., no toxicities grade >1 were observed [36].

Due to the preexisting risk for chronic kidney disease in VHL patients, our study put a special focus on kidney function in a long follow-up period. Renal function could be preserved in all patients, and no patient required dialysis. Furthermore, creatinine levels did not change considerably during the follow-up period, and levels at 2 years after SBRT were comparable to baseline levels.

Our findings were concordant with the results observed in other studies. In the pooled IROCK analysis including 223 patients, average eGFR decrease was 5.5 mL/min after SBRT [37]. In a prospective study of 21 patients treated with SBRT, the eGFR decreased on average by 8.7 mL/min at 1 year after treatment [28]. In the same study, split renal function decreased from 54.5% at baseline to 43.9% 1 year after SBRT. We found a smaller magnitude of decline in our cohort in which the split renal function of the kidney directly irradiated fell from 45.0% at baseline to 41.8% at 1 year after treatment.

The optimal number of fractions and total dose used for the treatment of RCC with SBRT is still unclear. In our study, we used a dose regimen of 50 Gy in 5 fractions without observing any relevant toxicity. A recent dose escalation study by Grubb et al. has shown that a dose of 60 Gy in 3 fractions could be safely used without any dose limiting toxicity [25]. Retrospective and prospective studies seem to demonstrate that ultra-hypofractionation or radiosurgery lead to better oncologic outcomes [36,37]. Further studies establishing the safety and efficacy of SBRT are ongoing. For example, the FASTRACK II trial (TROG 15.03, NCT02613819) is currently evaluating single fraction SBRT using 26 Gy versus 42 Gy in 3 fractions in a prospective manner [38].

Response evaluation of renal cancer after SBRT is still an unknown area. Like in other anatomic regions one of the challenges of SBRT is the interpretation of post-treatment radiographic studies. Response after SBRT is slow and often difficult to interpret. As RCC is generally slow growing, it has a similarly slow radiographic response to SBRT. The largest study of tumor response including 41 tumors treated with SBRT comes from a group of Harvard Medical School [39]. The mean pretreatment linear growth rate was 0.68 cm per year and the post-treatment linear growth rate regressed by an average of 0.37 cm per year. The study found no significant changes in enhancement on CT or MRI. This was also true for our study: All renal lesions decreased in size slowly but continuously for years after SBRT.

To the best of our knowledge, this is the first prospective study investigating the role of SBRT for ccRCC in VHL disease. Although the final conclusions are limited by the small sample size and non-randomized study design, it is a prospective study that gives insights in the feasibility and outcome of SBRT in VHL patients and can facilitate understanding of tumor response over a long follow-up interval in this patient population. Although our tertiary cancer centre has a specialized VHL outpatient clinic for patients with VHL disease, we only could include 7 patients in our prospective study, which is mainly related to the low prevalence of VHL disease.

Surgery is the standard curative-intent therapy for localized RCC. Other local non-surgical treatments, including SBRT, should only be considered in patients who have contraindications to surgery. However, as the treatment armamentarium grows, with more personalized potential treatment approaches becoming available, patient selection for a specific therapy becomes more and more important. For example, patients with bilateral renal tumors [40], and those with a contralateral recurrence after nephrectomy [41], could potentially be candidates for non-surgical local treatments. VHL disease patients who often develop bilateral renal tumors or multiple recurrences in the ipsilateral or contralateral kidney after nephrectomy, are deemed to be a patient cohort that might benefit from non-surgical local treatments, such as SBRT.

Treatment of early RCC lesions could provide a highly effective and well-tolerated therapy for hereditary renal cancer. In this disease highly prone to multiple recurrences, SBRT represents a novel, tissue-sparing treatment approach, to delay time to repeat surgery and premature renal failure.

## 5. Conclusions

In this prospective trial, SBRT for primary ccRCC in patients with VHL disease resulted in excellent LC and was well tolerated with long-term preservation of kidney function. Further prospective trials evaluating SBRT for localized RCC in VHL patients are required to investigate the role of SBRT as an alternative to surgery in this special patient population.

## Figures and Tables

**Figure 1 cancers-14-05069-f001:**
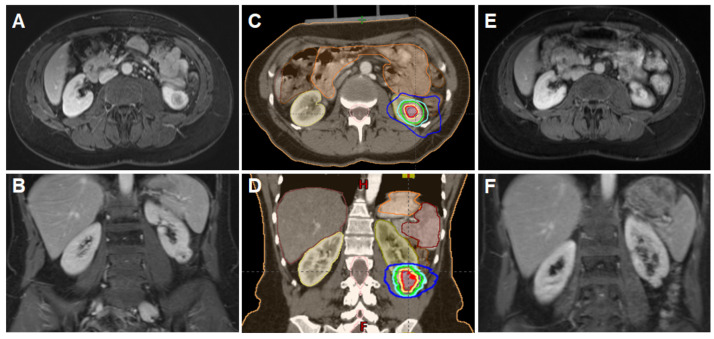
SBRT for ccRCC in a 45-year-old patient with VHL disease. A lesion (28 × 24 × 32 mm) was treated with SBRT (50 Gy in 5 fractions). (**A**,**B**) Axial and coronal MRI scans showing a lesion in the left kidney; (**C**,**D**) Planning CT with an axial and coronal scan demonstrating the treated lesion, several organs at risk (small bowel, large bowel, stomach, ipsi- and contralateral kidney) and different isodoses; (**E**,**F**) Axial and coronal MRI scans at 12-month follow-up showing tumor size reduction and central necrosis with no signs of local recurrence.

**Figure 2 cancers-14-05069-f002:**
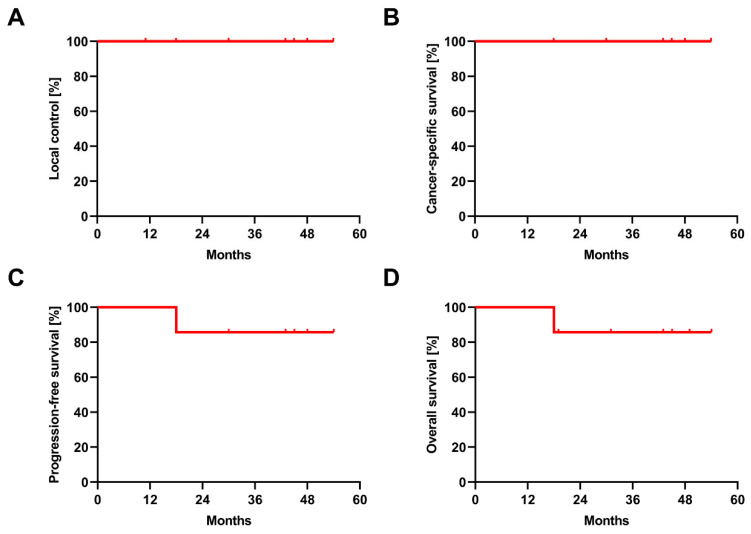
Oncological outcomes after SBRT in ccRCC. Kaplan–Meier curves for local control (**A**), cancer-specific survival (**B**), progression-free survival (**C**) and overall survival (**D**) after SBRT.

**Figure 3 cancers-14-05069-f003:**
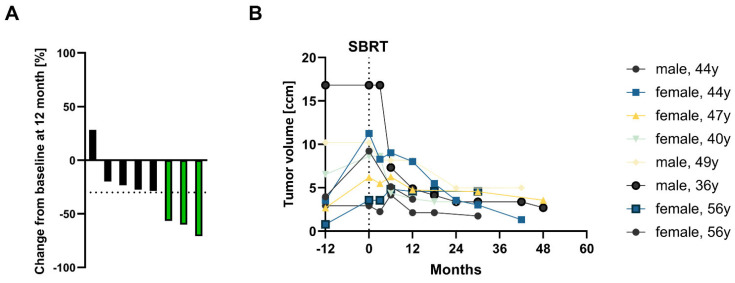
Relative and absolute tumor volume change before and after SBRT. (**A**) Relative change in tumor volume 12 months after SBRT (compared to baseline value at the start of SBRT) for individual lesions (*n* = 8). Bars in green indicate patients with partial remission defined by ≥30% decrease of tumor volume. (**B**) Growth kinetics of tumor volume 12 months prior to SBRT, immediately prior to SBRT and in the follow-up after SBRT (until the last follow-up MRI) for individual lesions.

**Figure 4 cancers-14-05069-f004:**
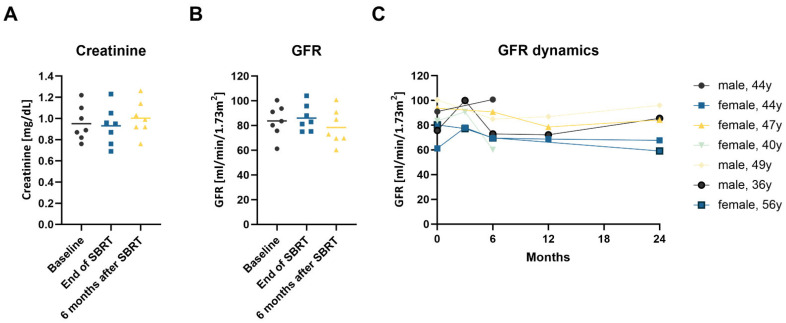
(**A**–**C**): Changes in kidney function after SBRT. (**A**) Creatinine serum concentration; (**B**) estimated GFR calculated following the CKD-EPI formula at baseline, immediately after SBRT and at the 6-month follow-up consultation. Each point represents one patient, and the line shows the mean value; (**C**) GFR dynamics in the first 24 months after SBRT for the individual patients.

**Table 1 cancers-14-05069-t001:** Patient and tumor characteristics of ccRCC patients with underlying VHL disease treated by SBRT.

		Median (Range)	
**Age at radiotherapy [years]**		44 (36–56)	
**Body Mass Index**		23.3 (20.3–25.8)	
		*n*	%
**Gender**	male	3	42.9
	female	4	57.1
**ECOG performance status**	0	4	57.1
	1	3	42.9
**Age-adjusted Charlson Comorbiditiy Index**	2	6	85.7
	3	1	14.3
**Prior kidney surgeries**	partial nephrectomy	3	42.9
	adrenalectomy	1	14.3
	none	3	42.9
**Kidney cysts**	multiple bilateral cysts	5	71.4
	solitary unilateral cyst	1	14.3
	no cysts	1	14.3
**Laterality**	left-sided	4	57.1
	right-sided	3	42.9
**Relative function of the treated kidney [%]**		49 (21–55)	
		**Median (range)**	
**Tumor size, largest diameter [cm]**		2.8 cm (1.9–3.5)	
**Renal Nephrometry Score**		8.5 (4–10)	
		** *n* **	%
**Renal Nephrometry Score (Complexity)**	low	2	25.0
	intermediate	4	50.0
	high	2	25.0

Function of the treated kidney relative to global kidney function was measured by renal nuclear scintigraphy with 102 MBq ^99m^Tc-MAG3.

**Table 2 cancers-14-05069-t002:** Treatment characteristics regarding SBRT for ccRCC in patients with underlying VHL disease.

		Median (Range)	
**Radiotherapy dose EQD2 (α/β = 10) [Gy]**		85.9 (83.3–87.5)	
**Radiotherapy dose BED [Gy]**		103.0 (103.0–123.8)	
**GTV [ccm]**		9.3 (3.5–14.3)	
**ITV [ccm]**		13.2 (5.6–22.1)	
**PTV [ccm]**		27.0 (15.6–44.0)	
		** *n* **	
**Fractionation**	5 × 10 Gy	7	
	8 × 7.5 Gy	1	

**Table 3 cancers-14-05069-t003:** Summary of treatment-related toxicities. Toxicities were classified according to CTCAE v4.0.

Acute		*n*	%
CTCAE grade 0		4	57.1
CTCAE grade 1		3	42.9
CTCAE grade 2–5		0	0
Chronic		*n*	%
CTCAE grade 0		7	0
CTCAE grade 1–5		0	0

## Data Availability

The data presented in this study are available on reasonable request from the corresponding author.

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
