# Peer review of "Stereotactic Body Radiotherapy for Renal Cell Carcinoma in Patients with Von Hippel–Lindau Disease—Results of a Prospective Trial"

_cancers, 2022, doi:10.3390/cancers14205069_

Round 1

Reviewer 1 Report

The authors report a study with 7 VHL patients treated with SBRT to improve local control.  They also look into kidney function with this treatment. 

Main limitations:  F/U, non-randomized, selection bias 

Author Response

We thank all reviewers for your valuable and affirmative comments for this study.

We absolutely agree with the mentioned limitations of the study. It was a non-randomized study introducing different bias that can be avoided by large randomized phase III study. The main limitations were mentioned in the discussion section. Median follow-up of 43 months was quite long compared to other studies of RFA or MWA.

Reviewer 2 Report

This is a very interesting pilot study opening the way to a new minimally invasive care of VHL.

Some points could be added:

-what were the exclusion criteria (tumor location, number of tumors, cystic lesions?)

-is it possible to add: the comorbidity Charlson index, the Body Mass Index of the patients?

-is-it possible to add the RENAL or PADUA or C-index tumor score?

-what was the median duration of one session

Author Response

We thank all reviewers for your valuable and constructive comments for this study. We have incorporated your suggestions into the manuscript, please see below.

Point 1: what were the exclusion criteria (tumor location, number of tumors, cystic lesions?

Response 1: Actually the only exclusion criteria was stage IV Chronic Kidney Disease. Tumor location or multiple cysts were not an exclusion criteria. Previous radiotherapy was added as an exclusion criteria.

Point 2: is it possible to add: the comorbidity Charlson index, the Body Mass Index of the patients?

Response 2: We included CCI and BMI in the methods and results section and table 1. Reference number 19 was added to the references.

Point 3: is-it possible to add the RENAL or PADUA or C-index tumor score?

Response 3: RENAL nephrometry score was added in the methods and results sections and in table 1. Reference number 20 was added to the references.

Point 4: what was the median duration of one session

Response 4: Median duration of one session was 20 minutes. This information was added to the text.

Reviewer 3 Report

Dear authors,

Thank you for your effort and endeavor.
 This is a study on pioneering issues that may intrigue any one working in the cancer field. Indeed, kidneys are tricky to treat. Conventionally fractionated radiotherapy did not show satisfactory results for renal cell carcinoma in the past. In contrast, the authors of present paper have revealed affirmative outcome from this stereotactic body radiotherapy (SBRT) study specifically focused on the patients with Von Hippel-Lindau disease (VHL), a hereditary disorder.

This paper is written by established researchers that are making an impact by sharing their work. The methods are clear and clarifying. It’s worth noting that the linac, tumor volume calculation method and treatment planning software they used were mentioned. One may be able to rely on rigorous and detailed description of their study process.

Author Response

We really appreciate your kind comments.

We are also convinced that SBRT of renal cancer is a highly effective treatment for local ablation of renal cell carcinoma. We were impressed that it is feasible in a patient cohort with multiple kidney lesions compromising kidney function. We are fully aware of that results are limited by the small patient number. Hopefully in the future bigger studies including more patients can be conducted.

Round 2

Reviewer 1 Report

ok